# Association between β2-Adrenoreceptor Medications and Risk of Parkinson’s Disease: A Meta-Analysis

**DOI:** 10.3390/medicina57101006

**Published:** 2021-09-24

**Authors:** Chu-Ling Chen, Shu-Yi Wang, Ta-Cheng Chen, Chieh-Sen Chuang

**Affiliations:** 1Department of Neurology, Yunlin Christian Hospital, Yunlin 648106, Taiwan; 820527@cch.org.tw; 2Department of Neurology, Changhua Christian Hospital, Changhua 50006, Taiwan; 67482@cch.org.tw; 3Division of Endocrinology and Metabolism, Department of Internal Medicine, Changhua Christian Hospital, Changhua 50006, Taiwan; 86761@cch.org.tw; 4College of Medicine, Kaohsiung Medical University, Kaohsiung 807378, Taiwan

**Keywords:** β2-adrenoreceptor, Parkinson’s disease, meta-analysis

## Abstract

*Background and Objective*: Parkinson’s disease (PD) is a progressive neurological disorder characterized by an accumulation of Lewy bodies and degeneration of dopaminergic neurons in the substantia nigra. The treatment options currently available are only partly effective and fail to restore the lost dopaminergic neurons or slow the progression. β2-adrenoceptors (β2AR) are widely expressed in various human tissues and organs, regulate many important metabolic functions, and are targeted for treatment of various diseases. Studies have reported a link between chronic use of the β2AR antagonist propranolol and an increased risk of PD, and chronic use of β2AR agonists has been associated with a decreased risk of PD. We conducted a meta-analysis on the association between both β2AR agonist level and β2AR antagonist level and the risk of PD. *Materials and Methods*: A comprehensive electronic search was conducted on the databases of PubMed, ScienceDirect, ProQuest, Cochrane Library, and ClinicalKey from the start of each database until 30 June 2021. The objective was to identify prospective cohort and case–control studies that have reported on the association between β-adrenoceptor agonist level, antagonist level, and PD risk. *Results*: A meta-analysis of the data extracted from eight studies revealed that β2AR agonist use was associated with reduced PD risk (RR = 0.859, 95% confidence interval [CI] 0.741–0.995. *p* = 0.043). Compared with the control group, β2AR antagonist use was associated with an increased risk of PD (RR = 1.490, 95% CI, 1.195 to 1.857. *p* < 0.005). Propranolol, a type of β2AR antagonist, was related to an increased risk of PD (RR = 2.820, 95% CI, 2.618 to 3.036. *p* < 0.005). *Conclusions*: In this meta-analysis, β2AR agonists were associated with a decreased risk of PD, and β2AR antagonists were related with an increased risk of PD. However, further studies with larger sample sizes and an evaluation of the long-term effects of varying dosages of medications are needed.

## 1. Introduction

Parkinson’s disease (PD) is the second most prevalent neurodegenerative disorder after dementia. However, PD is associated with more rapid progression towards disability and death [1]. PD is a chronic, progressive neurological disorder characterized by a loss of dopaminergic neuron and intracellular accumulation of Lewy bodies in the substantia nigra, resulting in tremor, bradykinesia, and rigidity [2,3]. A wide range of nonmotor symptoms occur in PD, including sleep disorder, cognitive impairment, depression, and autonomic dysfunction [4,5]. Olfactory dysfunction is widely acknowledged as one of the major nonmotor symptoms of PD, which often occurs before motor symptoms in PD [6,7]. Taste impairment has also been demonstrated in PD. It can start in the early stages of the disease but more frequently appears in advanced stages [8]. PD may lead to functional limitations through both motor and nonmotor symptoms. Therapeutic approaches focus on dopamine replacement and treating the disorder of motor function. Even though these medical treatments can initially ease symptoms, fluctuations in motor and nonmotor symptoms can develop over time, reducing the patient’s mobility and quality of life.

α-synuclein is a protein closely related to the occurrence of PD. When the protein is misfolded in nerve cells, Lewy bodies are formed, and excess accumulation can cause nerve cell damage [9,10]. The aggregation of the neuronal protein α-synuclein contributes to neuronal toxicity in PD. The accumulation of α-synuclein oligomers is currently a main therapeutic objective of candidate neuroprotective therapies [11]. A recent study suggested that α-synuclein can be regulated through modulation of β2-adrenoreceptors (β2AR) [12,13]. Several studies have established the neuroprotective effects of β2AR agonists in PD [12,14]. In addition, recent studies have shown that chronic use of β-blockers is related with an increased risk of PD [15,16]. However, some studies do not show a significant association between β2AR medications and risk of PD [10,17]. The aim of this meta-analysis was to evaluate the association between exposure to β2AR medications and risk of PD. 

## 2. Methods and Materials

This meta-analysis was conducted according to the Preferred Reporting Items for Systematic Reviews and Meta-Analyses (PRISMA) reporting guidelines (Figure 1) [18]. 

### 2.1. Literature Search and Screening

The literature was searched on PubMed, ScienceDirect, ProQuest, Cochrane Library, and ClinicalKey from the start of each database until 30 June 2021 using the following keywords: (Beta adrenoreceptor) and (Parkinson’s disease), (Beta Blocker) and (Parkinson’s disease), (β2-adrenoreceptor antagonist) and (Parkinson’s disease), and (β2-adrenoreceptor agonist) and (Parkinson’s disease). The search strategy included handsearching the reference lists of eligible studies and of recent reviews for additional records. Two independent reviewers screened potential studies by title and abstract to remove duplicate and obviously irrelevant studies. 

### 2.2. Data Extraction and Outcome Assessment

Two authors independently extracted data from the studies after verifying these studies’ eligibility for meta-analysis. The variables extracted from the studies were author names, year of publication, study sample size, sample age, gender, use of medications, follow-up duration, outcomes, and study quality evaluation. The outcome assessed in the meta-analysis was the risk ratio of PD in patients with exposure to β2AR agonist or antagonist. Studies were excluded if they were animal studies or studies not related to the treatment effect of PD.

### 2.3. Statistical Analysis

Due to the presumed heterogeneity of the included studies, the data were analyzed using random-effects meta-analysis models on Comprehensive Meta-Analysis software version 3 (Biostat, Englewood, NJ, USA). We calculated risk ratios (RRs) with a 95% confidence interval for outcomes. Publication bias was examined by visually inspecting funnel plots with the Egger’s regression test. The heterogeneity of the included studies was measured using a *Q* test and the *I*^2^ statistic. The threshold for statistical significance was set at an alpha level of 0.05 (two-tailed). 

## 3. Results

### 3.1. Search Results

Following the initial database search and screening, 56 full-text articles were assessed for eligibility. We ultimately included eight articles in this meta-analysis. Figure 1 illustrates the evaluation process we followed to select these studies. Table 1 lists the characteristics of all studies included in this meta-analysis. 

### 3.2. β2AR Agonist and Risk of PD

As indicated in Figure 2a, β2AR agonist treatment was more closely associated with the reduction in PD risk compared with non-β2AR-agonist treatment (RR = 0.859, 95% CI, 0.741 to 0.995. *p* = 0.043). Among these six studies, significant heterogeneity was noted (*Q* = 41.35, *I*^2^ = 87.9%, *p* < 0.05). The funnel plot of the included studies, for indicating publication bias, exhibited symmetry (Figure 2b). The Egger’s test revealed no publication bias (Intercept = −1.73, Standard error = 1.85, *p* = 0.40). A subgroup analysis for different types of β2AR agonist indicated that formoterol (RR = 0.824, 95% CI, 0.730 to 0.931. *p* = 0.002), salbutamol (RR = 0.888, 95% CI, 0.822 to 0.960. *p* = 0.004), salmeterol (RR = 0.931, 95% CI, 0.897 to 0.966. *p* < 0.005), and terbutaline (RR = 0.840, 95% CI, 0.714 to 0.987. *p* = 0.035) were related with a reduction in PD risk (Figure 2c). 

### 3.3. β2AR Antagonist and Risk of PD

The risk of PD was greater among patients who received β2AR antagonist treatment than among those in the control group (RR = 1.490, 95% CI, 1.195 to 1.857. *p* < 0.005) (Figure 3a). Among the seven studies, significant heterogeneity was found (Q = 96.23, *p* < 0.001, *I*^2^ = 93.8%). The funnel plot analysis for publication bias showed symmetry among the included studies (Figure 3b). Statistically, the Egger’s test showed no publication bias (Intercept = −3.34, Standard error = 3.27, *p* = 0.35). Propranolol, a type of β2AR antagonist, was often used to treat heart problems, help with anxiety, and prevent migraines. It was related with an increase in PD risk (RR = 2.820, 95% CI, 2.618 to 3.036. *p* < 0.005) (Figure 3c).

## 4. Discussion

In this meta-analysis, we found that β2AR agonists were related with a reduction in the risk of developing PD. For individual medications, formoterol, salbutamol, salmeterol, and terbutaline were associated with reduced risk of PD. By contrast, the use of β2AR antagonists, including propranolol, was related with an increased risk of PD.

The misfolding and aggregation of alpha synuclein protein, encoded by the SNCA gene, is an important cause of neuronal death in PD [21]. The overexpression of alpha synuclein could result in synaptic dysfunction, Golgi fragmentation, failure of chaperone mediated autophagy, and reduced complex I activity in mitochondria, which lead to neuronal cell death [14]. In human SK-N-MC neuroblastoma cells, β2AR agonists reduced SNCA mRNA abundance and alpha synuclein protein production. This study demonstrates that β2AR activators regulate SNCA transcription through the reduction of histone 3 lysine 27 (H3k27) acetylation, in which H3k27 is a promoter of the SNCA gene. By contrast, a β2AR antagonist, propranolol, increased the acetylation of H3k27 [12]. In a lipopolysaccharide (LPS) inflammatory PD rat model, β2AR stimulation reduced the motor deficits following intranigral LPS injection and decreased the loss of dopaminergic neuron and LPS-induced microgliosis in substantia nigra [22].

The association between exposure to β2AR agents and risk of PD may vary according to the presence of comorbidities. In a French nested case–control study, β2AR agonist exposure was associated with a reduced risk of PD only in the population without diabetes. Conversely, increased risk of PD was found in patients with diabetes who were exposed to a β2AR agonist [20]. Several studies have suggested that type 2 diabetes is associated with an increased risk of PD [23,24]. Because β-adrenoceptor agonists can influence glucose homeostasis by modulating insulin secretion, liver metabolism, and uptake of glucose into muscle [25], these results could be partly explained by poorly controlled diabetes, which would increase the risk of PD. 

A recent large case–control study was performed to a evaluate 31 potential modifiable protective factors or risk factors of PD, including lifestyle and environmental factors, and drugs and comorbid conditions [26]. The study enrolled 694 participants with PD and 640 healthy controls from six neurologic centers in Italy. This simultaneous assessment demonstrated that coffee consumption, smoking, and physical activity were independent protective factors for PD. Family history of PD, dyspepsia, and exposure to pesticides, oils, metals, and general anesthesia were independent risk factors for PD. The study indicates the need for potential preventive measures aimed at reducing the coexistence of multiple modifiable risk factors among patients with family history or advancing age.

This meta-analysis found that propranolol increased the risk of PD. However, the mechanism between exposure to β2AR blockers and risk of PD was unclear. The development of PD with propranolol use is considered a very rare adverse drug reaction at five-year follow-up (incidence < 1 case per 10,000 patients) [12,19]. Nevertheless, propranolol is widely used as a treatment for heart disease, migraine, and essential tremor, on which its positive effects are most apparent. Clinicians choose alternative treatment options or do not shoulder the risk after conducting a risk–benefit assessment. 

Our study has several limitations. First, PD is a neurological disorder with a complex pathogenesis implicating both environmental and genetic factors. For example, previous studies have reported that pesticide exposure and well water drinking increased the relative risk of PD [27,28]. These related environmental factors could not be assessed in these studies. Second, Cigarette smoke is a major cause of chronic obstructive pulmonary disease, which is frequently treated with β2AR agonists. Smoking cigarettes is associated with a decreased risk of developing PD according to population-based studies [29]. The effect of chronic nicotine use might be the relevant confounder. Furthermore, as in most meta-analyses, a potential limitation of the present study was the heterogeneity of the included studies in terms of their treatment duration, dosage of medications, and age of the participants.

## 5. Conclusions

The present meta-analysis shows evidence that β2AR agents are related to the risk of PD. However, future big data approaches or clinical trials with varying dosages and treatment durations will help us better assess PD risk and protective factors. 

## Figures and Tables

**Figure 1 medicina-57-01006-f001:**
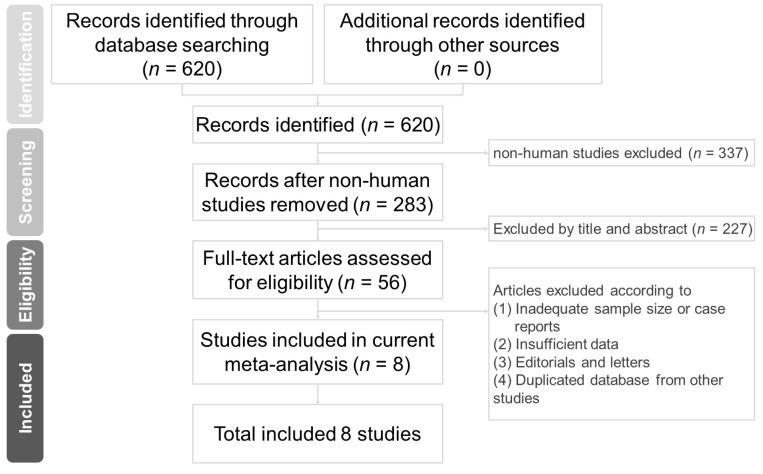
Flowchart of studies included and excluded.

**Figure 2 medicina-57-01006-f002:**
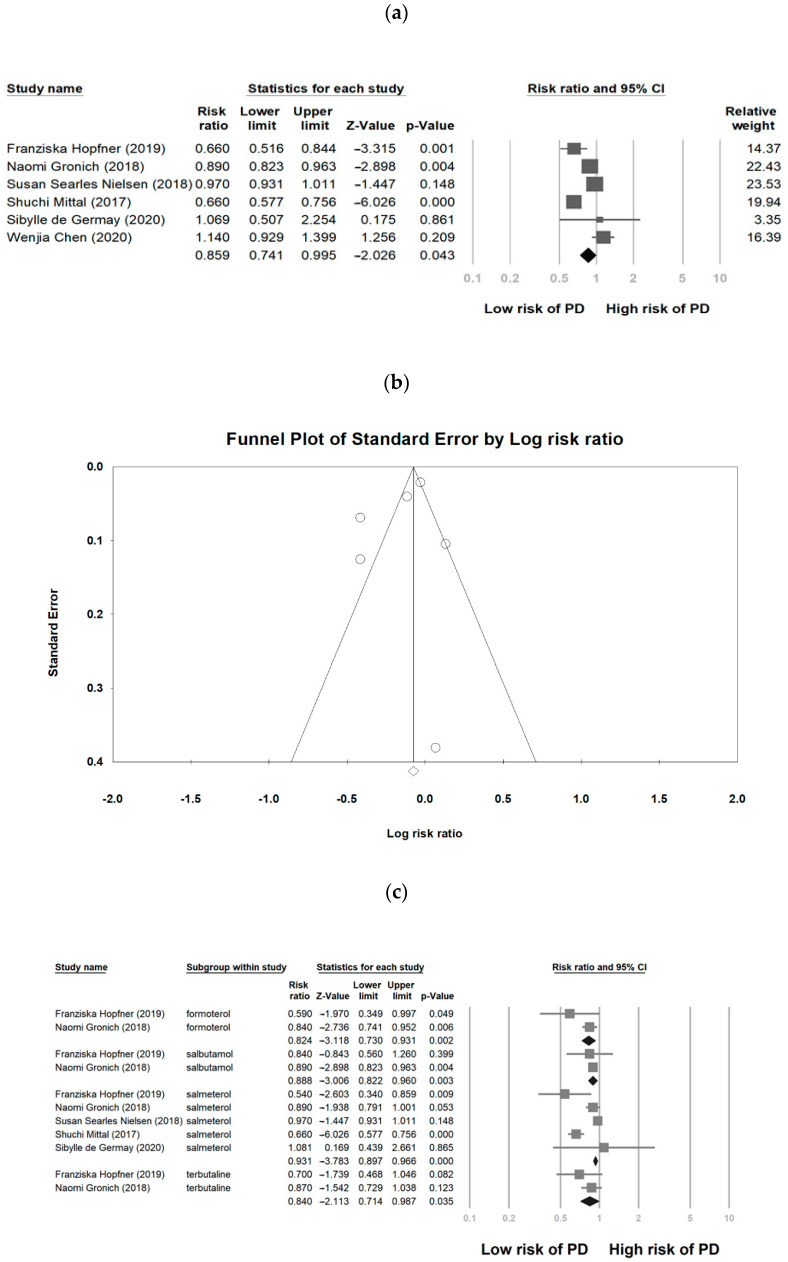
Forest plot of association between β2AR agonist and risk of PD (**a**), funnel plot analysis for publication bias (**b**), and subgroup analysis of different kinds of β2AR agonists and PD risk (**c**), PD: Parkinson’s disease, β2AR: β2-adrenoceptors.

**Figure 3 medicina-57-01006-f003:**
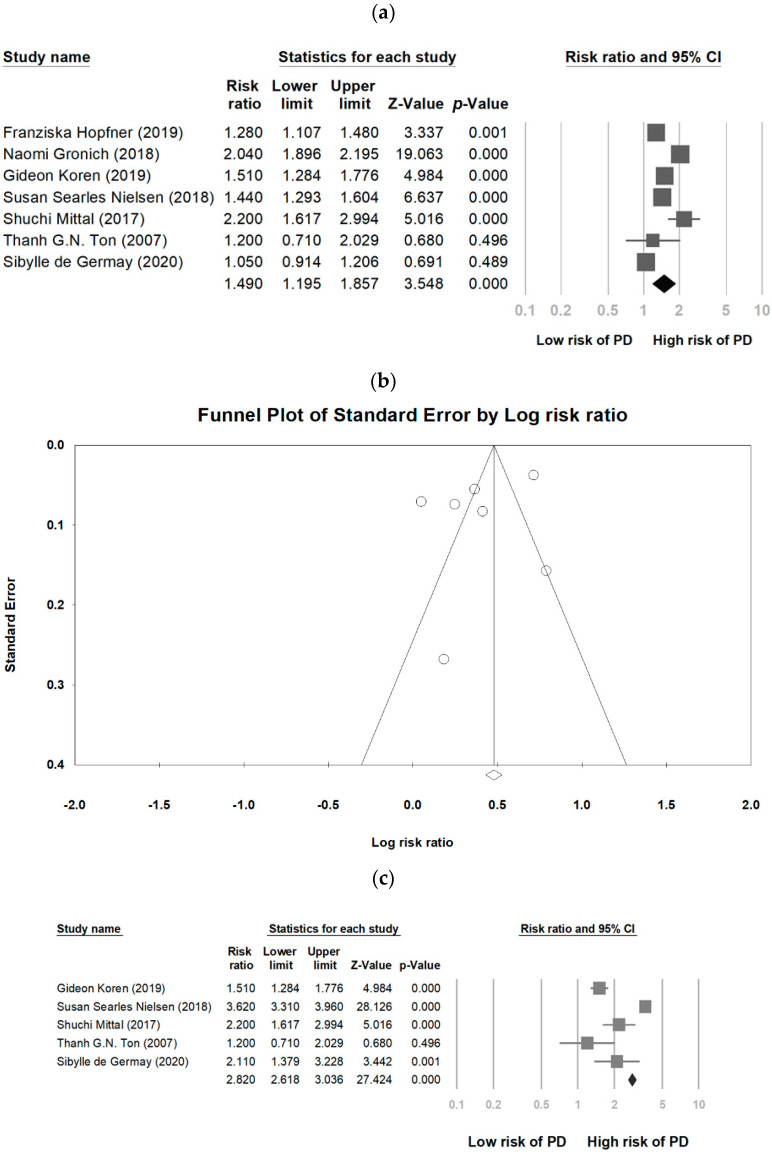
Forest plot of relations between β2AR antagonist and PD risk (**a**), funnel plot analysis for publication bias (**b**), and forest plot of the risk of PD with propranolol exposure (**c**).

**Table 1 medicina-57-01006-t001:** Summary of study characteristics in this meta-analysis.

Author (Year)	Medications	Study Type	Cases	Controls	Years of Follow-Up	Main Results
β2AR-agonist						
Shuchi Mittal (2017) [12]	salbutamol	prospective cohort	619,863	4,066,119	11	Salbutamol was associated with decreased risk of PD.
Susan Searles Nielsen (2018) [13]	Salbutamol	case–control	9514	8529	5	The positive association between salbutamol use and PD became non-significant after adjusting for cigarette smoking.
Naomi Gronich (2018) [16]	Salbutamol, Terbutaline, Salmeterol, Formoterol	case–control	1569	17,834	13	Use of β2-agonists was related with reduced risk of PD for short-acting, long-acting, and ultra-long-acting β2-agonists.
Franziska Hopfner (2019) [19]	Salbutamol, Terbutaline, Salmeterol, Formoterol	case–control	2790	11,160	>5	Use of β2AR agonist was related with reduced PD risk. The data indicated that the relation of β2AR agonists is indirectly mediated by smoking.
Sibylle de Germay (2020) [20]	Salbutamol	case–control	2225	2225	9	The relation between salbutamol exposure and PD was non-significant in diabetic or non-diabetic patients.
Wenjia Chen (2020) [10]	both short- and long-acting β2AR agonists	case–control	732	3660	>4	Use of β2AR agonists did not appear to affect the risk of PD in a COPD population.
β2AR-antagonist						
Thanh G.N. Ton (2007) [17]	Propranolol, Metoprolol, Atenolol, nadolol,	case–control	165	321	15	The data did not show a significant relation between PD risk and β-blockers.
Shuchi Mittal (2017) [12]	Propranolol	prospective cohort	9339	4,487,059	11	Individuals who used propranolol had a higher proportion of PD diagnosis during 11-year follow-up.
Susan Searles Nielsen (2018) [13]	propranolol, carvedilol, metoprolol	case–control	48,295	52,324	5	The results indicate that propranolol increases PD risk. The risk was nullified by adjusting for tremor and lagging propranolol exposure by 18 months.
Naomi Gronich (2018) [16]	nonselective β-ntagonists, selective β1-antagonists	case–control	4105	37,902	13	The use of β-antagonists was associated with an increased risk of PD. Increased risk of PD was seen with the use of nonselective β-antagonists, but not with the use of selective β1-antagonists.
Franziska Hopfner (2019) [19]	Propranolol, Metoprolol, Atenolol, Sotalol, Bisoprolol	case–control	407	1488	5	Increased PD risk was not found for all β2AR antagonists but only for metoprolol and propranolol.
Gideon Koren (2019) [15]	Selective and non- selective β-antagonists	prospective cohort	145,098	1,187,151	11	Chronic use of β-blockers confers a time- and dose-dependent increased risk for PD.
Sibylle de Germay (2020) [20]	Propranolol	case–control	595	561	9	A significantly positive association was found between increasing PD risk and propranolol.

PD: Parkinson’s disease; β2AR: β2-adrenoceptors; COPD: chronic obstructive pulmonary disease.

## Data Availability

The data supporting the findings of this study are available within the article.

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
