# Peer review of "Association between β2-Adrenoreceptor Medications and Risk of Parkinson’s Disease: A Meta-Analysis"

_medicina, 2021, doi:10.3390/medicina57101006_

Round 1

Reviewer 1 Report

It is of great interest a subject like the one presented in this study, as clinicians constantly need to check data for risk-benefit assessment of treatments.

Even if the number of studies included in the meta-analysis is not high at all, and rather small for a Funnel plot analysis, most of the studies are strong enough and include a very large sample and long follow-up period. Therefore, the present meta-analysis has a robust basis. What is more, the methodology followed is correct and the results are well presented.

Minor changes suggested:

# Lane23: “To the purpose” should be changed by “the objective”

# Lane23: this sentence is not clear as the term “Inception” seems to be referred to the start of each study. Related to this and the flowchart of studies, it is not clear what was the initial year for publication inclusion. Was any threshold set?

# Lane 95: Table1 is overall clear and well presented. However, third column header, “user”, should be changed to “case”, “patient”, or a more specific and clarifying term.

Reviewer 2 Report

This study is an interesting Meta-Analysis about the association between beta2-adrenoreceptors medications and PD risk. I have some comments to the authors:

1) Please correct in the abstract the typo "To the purpose"

2) In the introduction please state "quantitive and qualitative smell/taste disorders" rather than "hyposmia". Due to the increasing role of smell/taste function in  the understanding of PD, it's important to better expand the range of non-motor symptoms, with particular interest on smell/taste disturbances. Within this context, I invite the authors to add these three references related to the topic:

- Cecchini et al. Taste performance in Parkinson’s disease. J Neural Transm 2014;

- Solla et al. Frequency and Determinants of Olfactory Hallucinations in Parkinson's Disease Patients. Brain Sci 2021;

- Antje Haehner et al. Olfaction in Parkinson’s Disease – A Clinical Approach. European Neurological Review. 2020.

3) Please add in the Table a new column with the description of the study type (e.g. cross-sectional, prospective etc.). Also better define in the methods the inclusion criteria for the analysis.

4) Please add for each study the correspondent references number in the Table 1.

5) A recent paper published in the journal "Neurology" assessed several putative risk/protective factors of PD. Please add a comment on this in the discussion. 

- Belvisi et al. Risk factors of Parkinson disease: Simultaneous assessment, interactions, and etiologic subtypes. Neurology 2020.
